# Novel Therapeutic Approaches to Prevent Atherothrombotic Ischemic Stroke in Patients with Carotid Atherosclerosis

**DOI:** 10.3390/ijms241814325

**Published:** 2023-09-20

**Authors:** Núria Puig, Arnau Solé, Ana Aguilera-Simon, Raquel Griñán, Noemi Rotllan, Pol Camps-Renom, Sonia Benitez

**Affiliations:** 1Cardiovascular Biochemistry, Institut d’Investigació Biomèdica Sant Pau (IIB SANT PAU), 08041 Barcelona, Spain; npuigg@santpau.cat (N.P.); arnau-sole@hotmail.com (A.S.); 2Department of Biochemistry and Molecular Biology, Faculty of Medicine, Building M, Universitat Autònoma de Barcelona (UAB), Cerdanyola del Vallés, 08193 Barcelona, Spain; aaguileras@santpau.cat (A.A.-S.); rgrinan@santpau.cat (R.G.); 3Stroke Unit, Department of Neurology, Hospital de La Santa Creu i Sant Pau, Institut d’Investigació Biomèdica Sant Pau (IIB SANT PAU), 08041 Barcelona, Spain; 4Pathofisiology of Lipid-Related Deseases, Institut d’Investigació Biomèdica Sant Pau (IIB SANT PAU), 08041 Barcelona, Spain; nrotllanv@santpau.cat; 5CIBER of Diabetes and Metabolic Diseases (CIBERDEM), Instituto de Salud Carlos III, 28029 Madrid, Spain

**Keywords:** ischemic stroke, atherosclerosis, inflammation, therapies

## Abstract

Atherothrombotic stroke represents approximately 20% of all ischemic strokes. It is caused by large-artery atherosclerosis, mostly in the internal carotid artery, and it is associated with a high risk of early recurrence. After an ischemic stroke, tissue plasminogen activator is used in clinical practice, although it is not possible in all patients. In severe clinical situations, such as high carotid stenosis (≥70%), revascularization by carotid endarterectomy or by stent placement is carried out to avoid recurrences. In stroke prevention, the pharmacological recommendations are based on antithrombotic, lipid-lowering, and antihypertensive therapy. Inflammation is a promising target in stroke prevention, particularly in ischemic strokes associated with atherosclerosis. However, the use of anti-inflammatory strategies has been scarcely studied. No clinical trials are clearly successful and most preclinical studies are focused on protection after a stroke. The present review describes novel therapies addressed to counteract inflammation in the prevention of the first-ever or recurrent stroke. The putative clinical use of broad-spectrum and specific anti-inflammatory drugs, such as monoclonal antibodies and microRNAs (miRNAs) as regulators of atherosclerosis, will be outlined. Further studies are necessary to ascertain which patients may benefit from anti-inflammatory agents and how.

## 1. Introduction

Stroke is the second leading cause of death and the third cause of disability worldwide [1]. More than 20% of patients suffer a recurrence of stroke within 5 years, which increases the risk of severe disability [2]. Between 80–85% of all strokes are ischemic [3], and approximately 20% of them are caused by large-artery atherosclerosis [4], with atheromatous plaque in the internal carotid artery (ICA) being the most common. This type of ischemic stroke is named atherothrombotic stroke and shows a higher risk of recurrence than other stroke subtypes. Its pathogenesis is mainly related to atherosclerotic plaque rupture, which leads to thrombus formation with the clot blocking locally the blood vessel or provoking distal emboli, which ultimately interrupts oxygen supply to an area of the brain [5]. Otherwise, severe obstruction of the carotid or vertebral arteries by atherosclerosis may lead to hemodynamic alterations and hypoperfusion triggering an atherothrombotic stroke. In clinical practice, in addition to a healthy lifestyle, the current pharmacological treatments for stroke prevention are based on antithrombotic, lipid-lowering, hypoglycemic, and antihypertensive therapy [6]. In noncardioembolic ischemic stroke, lipid-lowering therapies, mainly high-dose statins, inhibitors of the 3-hydroxy-3-methyl-glutaryl-coenzyme A reductase (HMG-CoA), are essential to reduce low-density lipoprotein cholesterol (LDLc) levels to less than 70 mg/dL [7], as indicated by stroke guidelines [8]. In addition, treatment with high-dose statins in the acute phase of ischemic stroke and transient ischemic attack (TIA) reduces the NIHSS score and improves short-term functional outcomes without related adverse events [9]. Apart from statins, other lipid-lowering drugs (ezetimibe, proprotein convertase subtilisin/kexin type 9 (PCSK9) inhibitors) are recommended in a group of patients for stroke prevention. In antiplatelet therapy, aspirin, a nonsteroidal anti-inflammatory drug that inhibits cyclooxygenase, is the main treatment. Clopidogrel, an inhibitor of the P2Y12 receptor mainly found in platelets, and the combination of aspirin with dipyridamole, an inhibitor of phosphodiesterase 3 (PDE3), are also recommended. Recently, ticagrelor, another inhibitor of the P2Y12 receptor, has been also included in the stroke guidelines [10,11].

Atherothrombotic strokes due to symptomatic carotid plaques are associated with a threefold risk of early recurrence compared with the risk of recurrence in other stroke subtypes [12]. For this reason, apart from pharmacological treatment, revascularization may be considered in the setting of this stroke subtype. The principal revascularization techniques are carotid endarterectomy and plaque exclusion by stent placement. Although they are relatively safe and effective, current evidence supports revascularization only in some clinical situations, such as in severe carotid stenosis (70–99%) and moderate carotid stenosis (50–69%), mainly in men [13,14]. Nevertheless, despite the existing pharmacological and revascularization treatments, novel preventive therapies are needed to reduce the high number of recurrences in atherothrombotic stroke as well as the first event in high-risk asymptomatic patients with carotid atherosclerosis. 

Inflammation is a key factor in ischemic stroke and a potential target to prevent stroke in high-risk patients with carotid atherosclerosis. The use of therapies based on avoiding inflammation for the prevention of atherothrombotic stroke is the main topic of the present review. Indeed, medications currently used for stroke prevention, such as aspirin and antihypertensive drugs, have known anti-inflammatory effects [15]. In turn, statins exerted a dual lipid-lowering/anti-inflammatory role in the primary prevention of vascular events in the JUPITER trial [16], in which rosuvastatin reduced the risk of the first stroke by 48%. Finally, two new oral hypoglycemic agents named Semaglutide and Dulaglutide have been shown to reduce stroke incidence among diabetic patients, and additionally, semaglutide has been shown to reduce vascular inflammation in patients with atherosclerosis [17].

This comprehensive review not only discusses existing anti-inflammatory therapies but also highlights novel strategies addressed to counteract inflammation and their potential clinical use in the prevention of ischemic stroke, particularly of the atherothrombotic subtype, where inflammation plays a major role, as discussed in the next section. Hindering inflammation could slow the progression of the plaque to become vulnerable and, as a result, prevent the onset of the event. There are broad-spectrum well-known therapies, such as colchicine used long-term in low doses or others directed against specific inflammatory molecules by using monoclonal antibodies (MAb). The former often has multiple deleterious collateral effects, whereas the latter may fail due to the blocking of a single molecule since several interconnected molecules play a role in atherosclerosis. In this context, novel therapies targeting upstream mediators of inflammation may be of great interest. Unfortunately, there is a lack of knowledge on the use of novel anti-inflammatory therapies to prevent ischemic stroke, first-ever or recurrent stroke. Most studies are of a preclinical nature, with no clinical trials being clearly successful so far, and are usually focused on promoting protection after ischemic stroke. In this review, beyond lipid-lowering and antiplatelet drugs, the role of several inflammatory molecules as targets, including miRNA as regulators of atherosclerosis, in preventing ischemic stroke associated with carotid atherosclerosis is thoroughly discussed.

## 2. Inflammation in Atherothrombotic Ischemic Stroke

Inflammation has been proven to play a major role in the development and progression of atherosclerosis disease. In the earliest stages, LDL is retained within the subendothelial space where it is modified by oxidative stress and enzymatic activities, causing endothelial injury, abnormal lipid metabolism, and hemodynamic damage [18]. Modified LDL, which has inflammatory properties, induces the expression of adhesion molecules and chemokines, such as intercellular adhesion molecule-1 (ICAM-1), vascular adhesion molecule-1 (VCAM-1), E-selectin, P-selectin, monocyte chemoattractant protein-1 (MCP-1), and other inflammatory factors, in endothelial cells [19]. The presence of these inflammatory molecules causes lymphocytes and monocytes to infiltrate the arterial wall. The infiltrated monocytes are then differentiated into macrophages, which elicit a strong inflammatory response as a response to modified LDL by releasing more cytokines, adhesion molecules, and other molecules, such as tumor necrosis factor (TNF)-α and interleukin (IL)-1β and IL-6. In turn, the inflammatory microenvironment promotes the recruitment of vascular smooth muscle cells (VSMCs), which secrete connective tissue, contributing to the development of the fibrous cap within the atherosclerotic plaque [20]. In late-stage atherosclerosis, macrophages and VSMCs can take up modified LDL, becoming lipid-loaded foam cells. Necrotic macrophages secrete matrix metalloproteinases (MMPs) and other proteolytic enzymes that hydrolyze the extracellular matrix [20] and also release lipid, inflammatory, and prothrombotic molecules, leading eventually to plaque rupture, bleeding, and thrombosis, causing a cerebrovascular event in the case of carotid artery atherosclerosis [21].

A growing body of evidence supports the theory that inflammation and the immune system are key players in the pathophysiology of ischemic stroke, particularly when the origin is carotid atherosclerosis. Symptomatic carotid plaques show evidence of marked inflammation [22], including a high level of infiltration of monocyte/macrophage and T cells [23,24] in association with early recurrent stroke [25]. In addition, inflammation contributes to the brain damage caused by ischemia after the stroke. In turn, the damaged brain responds with a counteracting immunosuppressive effect that leads to the risk of infections and the release of mediators that induce tissue regeneration. Several inflammatory molecules participate in all the stages of the ischemic cascade, with a major role in the earlier stages, mainly in the formation of atherosclerotic plaque and its progression. 

Plaque formation and progression may be also influenced by arterial geometry and intraluminal hemodynamics. Some hemodynamics contribute to the shear stress and, thus, to chronic endothelial damage. In this regard, several studies have used computational models to analyze blood flow [26,27,28], with some of them demonstrating a relationship between arterial geometry and the risk of cardiovascular disease and stroke.

### 2.1. Cytokines

Several cytokines are known to play an essential role in the atherosclerotic process. TNF-α and interferon (INF)γ are type I cytokines produced by Th1 cells. In advanced phases of atherosclerosis, they induce effects leading to the narrowing and rupture of fibrous plaque. TNF-α promotes the expression of multiple pro-inflammatory genes inhibits anti-atherogenic genes and is involved in foam cell formation [29]. After ischemic stroke, it induces ICAM-1 and increases the expression of MMPs, promoting blood–brain barrier (BBB) disruption [30]. IFN-γ is involved in immune cell recruitment, LDL accumulation, inflammation, plaque development, and stabilization [31]. A study suggested that IFN-γ and T cells could be considered therapeutic targets for ischemic stroke in the late phase of ischemic stroke [32].

IL-1β is secreted as a propeptide that needs to be cleaved by caspase 1 to perform its functions. IL-1β promotes the recruitment of leukocytes to the atherosclerotic plaque and also to the ischemic brain region extending the cerebral infarct area [33]. IL-1β is considered a pro-inflammatory cytokine whose effects depend on the haplotype, with some of them linked to the risk of stroke. Among them, haplotype 4 is associated with the greatest risk [34].

IL-6 is a pro-inflammatory cytokine secreted by different cell types that are involved in several processes including the activation of endothelial cells, differentiation and oxidation processes of lipoproteins, the stimulation of hepatic synthesis of acute phase reactants, high-sensitivity C-reactive protein (hsCRP) and fibrinogen, leukocyte recruitment and the stimulation of lymphocyte proliferation [35,36]. As a result, IL-6 accelerates the development of atherosclerotic lesions [37,38]. On the other hand, IL-6 has also been shown to have some anti-atherogenic properties [39]. 

hsCRP has been reported as a marker of inflammation associated with an increased risk of the first ischemic stroke [40]. Indeed, the Emerging Risk Factors Collaboration, in an analysis of over 160,000 patients, reported a linear relationship between hsCRP and the first stroke and coronary events [41]. Some data also point to an association of hsCRP and IL-6 with recurrent stroke [42]. 

Some anti-atherogenic cytokines are transforming growth factor-β (TGF-β), which inhibits cell MMPs, and anti-inflammatory IL-10. After ischemia, TGF-β1 is induced to regulate tissue damage and promote repair [43]. IL-10 is closely related to the prevention of the progress of atherosclerosis [44] and seems to play an immunosuppressive role after a stroke, which could otherwise increase the risk of bacterial infection [45].

Chemokines, such as MCP-1 and IL-8, are involved in leukocyte recruitment to the lesion area. MCP-1, also known as CCL2, regulates the migration and infiltration of monocytes, T lymphocytes, and natural killer cells [46]. Increased levels of MCP-1 in the blood have been observed in patients with ischemic stroke compared with the control group, independently of known risk factors [47,48]. IL-8 induces the expression of integrins in neutrophils, promoting their adhesion to the endothelium [49]. A study showed that after ischemic stroke, high levels of IL-8 were observed [50].

### 2.2. MMPs

MMPs constitute a family of endopeptidases, zinc-containing and calcium-dependent enzymes that are mainly released by foam cells with the function of degrading proteins of the extracellular matrix (collagen, elastin, fibronectin, and others), which give stability to atherosclerotic plaque [51]. Accordingly, MMPs are higher in unstable plaques [52,53]. Thus, it would be therapeutically interesting to decrease the activity of these enzymes to decrease the risk of plaque rupture. However, this approach is controversial since MMPs also play a beneficial role in remodeling, which may hinder the usefulness of inhibiting these enzymes [54].

MMPs with collagenase action have been associated with plaque destabilization [55]. Among them, MMP-8 has proven to exert a strong effect [56]. Indeed, the presence of MMP-8 together with MMP-12 in macrophages within the plaque was associated with an increased risk of major cardiovascular (CV) events and stroke [57,58]. MMPs with gelatinase activity, such as MMP-2 and MMP-9, have been in the spotlight of many CV diseases, as they highly induce VSMC migration and proliferation [59]. Some approaches, such as MMP-9 gene silencing [60], have been shown to avoid the deleterious effects of MMP-9. 

MMP-3 and MMP-7 are also related to carotid atherosclerosis and stroke [61], with some polymorphisms causing variability in the risk of suffering from stroke [62,63,64]. MMP-14 is upregulated during human atherosclerotic plaque progression and in symptomatic carotid plaques. Moreover, it plays an important role in plaque neovascularization and the activation of other MMPs [65,66]. 

Tissue inhibitors of MMPs (TIMPs) have been suggested as a therapeutic tool for delaying the development of atherosclerosis, such as the potent inhibitor of MMP-9. In this regard, TIMP-3, which inhibits several MMPs [54,67], is a candidate for therapeutic use.

### 2.3. Adhesion Molecules

Cell adhesion molecules (CAMs) are located on cell surfaces and are involved in binding to other cells or to the extracellular matrix. In atherosclerosis, several CAMs expressed on the endothelial cell membrane are involved in leukocyte recruitment, enhancing the number of inflammatory cells within the lesion [68]. E-selectin promotes the initial binding and rolling of inflammatory cells by interacting with carbohydrates present on the surface of leukocytes. Then, a tight attachment is allowed by the interaction of the CAMs of the immunoglobulin superfamily ICAM-1 and VCAM-1 with their counterparts in leukocytes. The chemokine fractalkine (FKN), when expressed on the membrane of activated endothelial cells, also shows the ability to tightly adhere to leukocytes [69]. 

CAMs play an outstanding role in ischemic stroke, as demonstrated in animal models [70]. ICAM-1 is highly expressed in atherosclerotic carotid plaques of symptomatic patients compared to asymptomatic plaques [71]. In this context, targeting CAMs and, thus, inhibiting the binding of leukocytes to endothelium may slow the development of atherosclerotic plaque, which would be useful for the prevention of ischemic strokes in patients with carotid atherosclerosis. In a post-ischemic situation, experimental studies revealed that the injection of antibodies against ICAM-1 in rats reduced infarct size after transient unilateral stroke [72]. However, the murine antihuman ICAM-1 MAb (enlimomab) treatment did not show efficacy in a human clinical trial conducted in ischemic stroke patients [73], as discussed below. 

In a highly inflammatory microenvironment, as is that of advanced atherosclerotic plaque, activated cells express not only increased levels of membrane-bound adhesion molecules but also release their soluble forms into circulation [74]. The shedding of CAM from the membrane is an active process regulated by proteolytic enzyme activity, mainly MMPs [75]. The function of these soluble forms in atherosclerosis and ischemic stroke remains to be elucidated. On the one hand, they may avoid the recruitment of more inflammatory cells to the lesion; on the other, they are known to have some deleterious effects, such as the secretion of inflammatory cytokines. 

The soluble(s) forms of selectins, including sE-selectin and sP-selectin, are elevated in ischemic stroke [76]. In the atherothrombotic subtype, sP-selectin levels are elevated in the acute phase [77] and they are associated with sE-selectin levels in the subacute phase [78]. As well as selectins, concentrations of sICAM-1 and sVCAM-1 are increased in ischemic stroke patients [70,79,80]. Moreover, serum sICAM-1 concentrations at admission are predictors of the prognosis of ischemic stroke patients [81]. The plasma concentration of sICAM-1 has been proposed as a marker of early atherosclerosis and of subclinical coronary disease in humans [82,83] and of the progression of atherosclerosis in mice [84]. As ICAM-1 is a molecule expressed not only in endothelial cells but also in macrophages, the released soluble form may indicate both endothelial dysfunction and macrophage activation. A recent study has revealed that in ischemic stroke patients, sICAM-1, sVCAM-1, and FKN were associated with carotid plaque inflammation, measured by the gold-standard method ^18^F-fluorodeoxyglucose positron emission tomography (^18^F-FDG PET) [80]. Of these soluble CAMs, only sICAM-1 predicted ischemic stroke recurrence and was associated with the presence of highly inflamed carotid plaques with high sensitivity, thereby suggesting that sICAM-1 mirrors the inflammatory state of the atherosclerotic plaque and is a feasible target to prevent the progression of the lesion and the ensuing ischemic event.

### 2.4. Cell and Soluble Forms of Receptors

Some cell receptors, as well as their soluble forms, related to cell inflammation and internalization of modified lipoproteins, are increased in ischemic stroke patients, thus being potential candidates for suppression and developing a strategy to hamper the onset of stroke events. Once again, it is important to highlight that this would be of particular importance in the subtype associated with atherosclerosis, in which lipid accumulation and inflammation are the main inductors.

Lectin-like oxidized LDL receptor-1 (LOX-1) is a scavenger receptor found in atherosclerotic carotid lesions whose expression within the carotid plaques is increased versus non-atherosclerotic vessels [85]. The shedding of LOX-1 can be elicited by inflammatory molecules and by oxidized LDL (oxLDL). sLOX-1 is then released into the systemic circulation, especially in ruptured atherosclerotic lesions [86]. Several studies have found that sLOX-1 concentration is increased in ischemic stroke patients [87,88] and in asymptomatic patients at high risk of stroke [89].

The soluble forms of other receptors related to inflammation and foam cell formation, such as the cluster of differentiation (CD) 163, CD36, CD14, and CD63, and low-density lipoprotein receptor-related protein 1 (LRP1), have been proposed as biomarkers for ischemic stroke, particularly for the atherothrombotic subtype [90,91,92,93]. In carotid plaque, the expression levels of CD36 and CD163 are higher in vulnerable plaques and symptomatic plaques [90,94].

sLRP1 had been postulated as being a predictive biomarker for coronary artery disease (CAD) risk [95]. A recent study revealed that it was also independently associated with the degree of carotid plaque inflammation measured by ^18^F-FGD PET in ischemic stroke patients and predicted highly inflamed plaques with high sensitivity [96]. These results suggest that sLRP1 is a surrogate marker of carotid plaque inflammation that may be targeted in future therapeutic strategies.

## 3. Biomarkers of Disease Progression

Carotid atherosclerosis can be caused by risk factors like hypertension, diabetes, obesity, smoking, and genetic predisposition due to their effect on LDL particles and inflammation. Progression of carotid atherosclerosis is related to a higher risk of vascular events compared with atherosclerosis, which remains stable over time. Unstable plaques are characterized by showing a necrotic core with an overlying thin/ruptured cap, and strong intraplaque inflammatory processes, such as endothelial dysfunction, macrophage activation, oxidative stress, lipid deposition, and neovascularization. Unstable/vulnerable plaques can be detected by imaging techniques, owing to fast progression in the degree of stenosis and an echolucent appearance on ultrasonography [97]. 

Surrogate markers of progression are necessary for obtaining information and monitoring the disease before an ischemic stroke event occurs. These markers could help us to prevent the disease and could lead to personalized medicines and the design of preclinical assays to identify new therapies. Among the potential biomarkers of carotid plaque progression, inflammatory and lipid biomarkers are briefly described below [98].

Among inflammatory markers, IL6 is the main candidate to be considered as an indicator of plaque progression, whereas there is a lack of evidence concerning other cytokines and the risk of atherosclerosis progression in patients with carotid stenosis or ischemic stroke. The first population-based study demonstrating the role of IL-6 as an independent predictor of plaque progression in atherosclerosis disease was the Tromso study [99], which was conducted with a relatively small sample size. However, this study did not reveal an association between circulating levels of IL-6 and plaque severity or vulnerability. Nevertheless, the relationship with plaque progression was supported by a recent and noteworthy analysis of a large population-based Cardiovascular Health Study (CHS), which demonstrated that circulating IL-6 predicts carotid plaque severity and vulnerability as well as plaque progression at 5 years [100]. This study demonstrated the relationship between levels of IL-6 and high-risk plaque features associated with stroke risk. The authors identified a cutoff point (2.0 pg/mL) as a threshold for selecting patients who would benefit from anti-IL-6 drugs for stroke prevention.

Although lipid markers are not the focus of this review, their relationship with plaque vulnerability and progression deserves to be briefly discussed. High-density lipoprotein (HDL) is a lipid biomarker with properties conferring atheroprotection, such as promoting reverse cholesterol transport and inhibiting lipoprotein oxidation [101]. Several studies have demonstrated the inverse relationship between HDL cholesterol and carotid atherosclerosis [102,103]. The Tromso study showed for the first time that high levels of HDL cholesterol are inversely associated with plaque growth [104].

OxLDL is formed as a result of oxidative stress in the arterial wall and is a well-known factor in the development of atherosclerosis by inducing inflammation and foam cell formation [105,106]. Nishi K et al. described that high plasma and plaque oxLDL are associated with plaque vulnerability [105], which was corroborated later, as well as the association with the stroke outcome [107]. A recent study has reported that plasma oxLDL levels were increased in ischemic stroke patients with carotid atherosclerosis, with no association with imaging features of carotid vulnerability. However, electronegative LDL (LDL(−)), a group of heterogeneous modified LDLs with increased negative charge and inflammatory properties, was increased in those patients in association with the degree of stenosis and with the presence of hypoechoic plaque and intraplaque neovessels [108]. 

Lipoprotein-associated phospholipase A2 (Lp-PLA2) is an enzyme associated with lipoproteins and, within LDL particles, with LDL(−) [109]. It has been expressed in the necrotic center of the atherosclerotic plaques in macrophage-rich areas [110]. In a small sub-study of the NASCET trial, Lp-LPA2 was increased in patients with high-grade carotid stenosis and unstable plaque [111].

## 4. Pharmacological Therapies Based on Inflammatory Molecules

Besides antithrombotic therapies, the usual interventions to slow the progression of atherosclerosis have been the control of vascular risk factors and particularly the reduction in plasma cholesterol levels. In this regard, it has been shown that every 1 mmol/L reduction in LDLc is associated with a 20% relative reduction in future CV events. In addition, several drugs aimed at counteracting an excessive inflammatory response have been recently proposed as potential therapies for atherosclerosis and atherothrombotic ischemic stroke. Some of them have been tested in clinical trials that suggested that aggressive inhibition of inflammation may be a crucial therapeutic target for secondary prevention in high-risk patients—a topic that has been covered in a few previous reviews [15,112]. The main anti-inflammatory strategies, which will be discussed throughout the present review, based on slowing the progression of atherosclerosis to prevent ischemic stroke are summarized in Figure 1.

Some new stroke anti-inflammatory approaches based on studies in animal models inhibit the acute innate immune response (to limit excessive damage in the ischemic brain) and the post-acute modulation of the adaptive immune system (to limit post-stroke complications). Antileukocyte strategies, including the blocking of adhesion molecules, have been demonstrated to reduce ischemic brain injury in animal models. These treatments showed longer therapeutic windows (12–24 h after stroke) than tPA and may be administered together with reperfusion therapy [113,114]. These therapies could also be beneficial when hemorrhagic stroke has not been discarded [115]. Specifically, ApTOLL, an aptamer that antagonizes Toll-like receptor 4 (TLR4), was safe and associated with a reduction in mortality and disability at 90 days in patients with ischemic stroke in Phase I/II randomized clinical trial when administered within 6 h of onset in combination with endovascular treatment [116]. Interestingly, this kind of anti-inflammatory therapies could also be useful for preventing the development of atherosclerotic plaques to avoid a first event or an ischemic stroke recurrence, particularly in patients with high-risk carotid plaques who are not eligible for surgical revascularization [117]. 

A first line of anti-inflammatory therapy is the long-term use of broad-spectrum agents, mainly low doses of colchicine or methotrexate. However, these agents usually have undesirable side effects, such as the renal toxicity ascribed to colchicine. On the other hand, studies based on inhibiting specific cytokines have shown promising but inconclusive results. Inflammatory cytokines can be blocked by different approaches, such as the use of soluble receptors and MAb against specific inflammatory mediators, such as IL-1β and IL-6. CANTOS trial reported the beneficial effect on CV risk of blocking IL-1β [118]. Unfortunately, trials assessing anti-inflammatory agents have used composite endpoints of CV risk and not specifically stroke caused by carotid atherosclerosis. Other trials evaluating agents that inhibit TNF-α, CAMs, leukotrienes, secretory phospholipases, and inflammation-associated antioxidants have been ineffective for event reduction [119]. Novel anti-inflammatory and anti-cytokine agents targeting molecules upstream of IL-1β, such as NLR family pyrin domain-containing 3 (NLRP3) inhibition and other pathways of innate immunity, have also been proposed [120]. Clinical trials to date involving anti-inflammatory therapies that include stroke among clinical outcomes are summarized in Table 1.

The conflicting results regarding anti-inflammatory therapy on vascular diseases and particularly on stroke prevention suggest the need for future trials in which the results should be analyzed by stroke subtypes and the use of anti-inflammatory therapies, coupled with other approaches, including cholesterol-lowering treatment and statin therapy [119].

### 4.1. Broad-Spectrum Anti-Inflammatory Drugs

#### 4.1.1. Colchicine

Colchicine is an anti-inflammatory remedy that has been used for centuries in the prevention of inflammatory diseases. Colchicine binds to tubulin, thereby altering its conformation. As a result, the assembly of the inflammasome is altered, leading to decreased levels of interleukins IL-1β and IL-18 [121]. As colchicine decreases both L-selectin and E-selectin expression, neutrophil recruitment to endothelium is diminished as well as the ensuing deleterious actions, such as the release of superoxide and neutrophil extracellular traps. Colchicine also prevents platelet aggregation. Therefore, it has broader anti-inflammatory effects beyond inhibition of IL-1β and is a strong candidate as a damper of general inflammation, exerting beneficial effects in CV medicine. To date, independent randomized controlled trials evaluating the effect of long-term low-dose colchicine in a broad number of patients with acute and chronic coronary disease demonstrated that colchicine may reduce the risk of CV death, myocardial infarction, ischemic stroke, and ischemia-driven revascularization.

First, Nidorf et al., demonstrated that colchicine had anti-inflammatory effects in addition to those of statin and antiplatelet therapy in patients with stable coronary disease [122]. In a pilot study (LoDoCo) conducted in patients with stable coronary disease, colchicine decreased the occurrence of unstable angina [123]. This observation was corroborated in a larger cohort (LoDoCo2) in which colchicine diminished CV death, myocardial infarction, ischemic stroke, and ischemia-driven coronary revascularization [124]. However, in acute coronary syndrome, colchicine has shown mixed results [121]. The COLCOT trial powerfully assessed the clinical effects of colchicine following myocardial infarction [125]. In this trial, patients treated with 0.5 mg colchicine daily had a lower incidence of CV death, cardiac arrest, myocardial infarction, and, especially, stroke. COLCHICINE-PCI trial showed that the preprocedural administration of colchicine decreased IL-6 and hsCRP when compared with placebo but did not affect enzymatic measures of infarct size [126]. In the COPS trial, the colchicine group showed an improved clinical outcome (including less noncardioembolic ischemic stroke) than the placebo group [127].

Overall, the above-mentioned independent randomized clinical trials conducted in >11,000 patients with acute and chronic CV disease, followed for up to 5 years, demonstrated that colchicine safely slows the progression of atherosclerosis and reduces the risk of CV disease. However, they have some limitations, such as the lack of clinical or biological markers for the selection of participants or no data regarding lipid levels or blood pressure at enrollment. Moreover, some patients showed intolerance to colchicine (10% in the COPS trial) and, in LoDoCo 2 and COPS trials, there was a higher incidence of non-CV death in patients receiving colchicine, although without finding a direct association. 

Other ongoing studies will provide further information about the use of colchicine in subsets of patients with CV disease, such as the CLEAR SYNERGY study, the COLCARDIO trial (ACTRN12616000400460), and the CONVINCE trial (NCT02898610) [121], a Phase 3 trial conducted in non-severe ischemic stroke (noncardioembolic) that evaluates the recurrence of vascular events [128]. CASPER trial will investigate colchicine for secondary prevention after a stroke, in patients with elevated hsCRP. Finally, the CIAFS-1 pilot trial is investigating the effect of colchicine on reducing markers of inflammation and thrombosis in anticoagulated atrial fibrillation patients in preparation for a Phase 3 trial for the prevention of stroke and systemic embolism.

#### 4.1.2. Other Broad-Spectrum Anti-Inflammatory Treatments

Besides colchicine, other anti-inflammatory treatments have been proposed for ischemic stroke with mixed results. In the CIRT trial, a low dose of the nonspecific anti-inflammatory agent methotrexate failed to reduce recurrent vascular events in patients with coronary disease [129]. Minocycline has been tested in clinical trials showing broad anti-inflammatory and neuroprotective properties [130]. Vinpocetine, an anti-inflammatory alkaloid that diminishes the release of inflammatory cytokines and chemokines through nuclear factor kappa-light-chain-enhancer of activated B cells (NF-κB) inhibition has shown an anti-inflammatory role in atherosclerosis and early inflammation associated with an ischemic stroke [131]. Melatonin is another agent with potent anti-inflammatory, antioxidative, and neuroprotective properties. Some studies have pointed to beneficial effects on carotid artery stenosis by diminishing endothelial damage, stabilizing arterial plaque, and reducing cerebral ischemia/reperfusion injury [132].

### 4.2. Specific Anti-Inflammatory Drugs

The main specific targets proposed for stroke therapies are the cytokines TNF-α, IL-1β, and IL-6, and the adhesion molecule ICAM-1, which will be discussed in this section. Other molecule tributaries to be targeted are MMPs, including SB-3CT, a gelatinase inhibitor and ADAM17 inhibitor [133], and other specific MMP inhibitors (BB-94, BB-1101, GM6001, and FN-439).

Some trials have targeted Lp-PLA2 by using specific antagonists (darapladib, varespladib) but without showing benefit in patients with recent or stable coronary disease [134].

#### 4.2.1. TNF-α

TNF-α is targeted by drugs such as infliximab, etanercept, and adalimumab [134]. TNF-α inhibition affects downstream inflammatory cytokines such as IL-6. Thus, TNF-α antagonists may inhibit early stages of atherosclerosis development by downregulating the expression of VCAM-1 and ensuing smooth muscle proliferation and adherence of leucocytes to endothelial cells. To date, no clinical trials have assessed the potential benefit for stroke prevention of TNF-α antagonists in patients with CV diseases.

#### 4.2.2. IL-1β

The CANTOS trial focused on the effect of blocking IL-1β by the MAb canakinumab in CV disease. The patients in this study had stable atherosclerosis and were on statin therapy but with residual inflammatory risk. High doses of canakinumab promoted a 15% reduction in major adverse CV events and a 17% reduction in major adverse CV events plus urgent revascularization in comparison with placebo [118] and a high reduction in plasma levels of the marker of inflammation hsCRP (concentrations of less than 2 mg/L) [135]. These effects were independent of lipid-lowering and blood pressure control. However, a subgroup analysis of the CANTOS trial did not show benefits for stroke prevention [118]. This lack of effect in stroke may be because of non-atherothrombotic causes of stroke. 

Agents proposed for counteracting the action of IL1-β, although not currently approved for vascular treatment, include recombinant IL-1 receptor antagonist (anakinra), sIL-1 receptor chimeric fusion protein (rilonacept), and oral NLRP3 inflammasome inhibitors, which inhibit the formation of active IL-1β [136]. Anakinra reduces the release of hsCRP, but this compound seems to lead to a dual IL-1α and IL-1β inhibition and may therefore not be optimal for atheroprotection or providing the best safety balance between IL-1 activation and inhibition [137].

#### 4.2.3. IL-6

The recent and noteworthy study of Kamtchum-Tatuene et al. demonstrated the relationship between levels of IL-6 and high-risk plaque features associated with stroke risk [100]. The authors identified a cutoff point (2 pg/mL) as a threshold for selecting patients who would benefit from anti-IL-6 drugs for stroke prevention. Targeting IL-6 with agents may cause both membrane-bound and circulating IL-6 receptors to be blocked. This may help to minimize endothelial dysfunction and the pro-inflammatory effect of IL-6, being useful for not only atherothrombotic stroke but atherosclerosis prevention as well.

The CANTOS trial helped define the inflammatory pathway from IL-1β to IL-6 to hsCRP as a central target for atheroprotection. Indeed, further analyses showed that the effect of canakinumab was mediated by the reduction in circulating levels of IL-6 [138,139], an observation that is supported by genetic studies [140,141]. Together, this evidence suggests that targeting IL-6 with MAb may be an adjuvant treatment for preventing ischemic stroke in patients with carotid atherosclerosis.

Ziltivekimab is an IL-6 ligand MAb that was developed specifically for atherosclerosis in patients with chronic kidney disease [142]. In the RESCUE trial, (hsCRP), Lp-PLA2, and lipoprotein (a) (Lp(a)) levels were reduced in patients with ziltivekimab, whereas HDL cholesterol was not affected [143]. Following on from these results, an ongoing trial, ZEUS, will compare ziltivekimab with placebo in patients with chronic kidney disease and elevated hsCRP to determine if the anti-inflammatory approach of directly reducing circulating IL-6 reduces CV event rates [142].

Tocilizumab, an IL-6 receptor antibody, has shown conflicting results regarding CV disease prevention. In patients with rheumatoid arthritis, tocilizumab decreased hepatic LDL receptor expression [144] and induced elevations in LDL cholesterol, as well as promoted anti-inflammatory properties in HDL [145], improved endothelial dysfunction [146], and decreased Lp(a) levels [147].

Again, to date, no clinical trials have addressed directly the potential benefit of IL-6 inhibitors for stroke prevention.

#### 4.2.4. ICAM-1

The recruitment of leukocytes to the endothelium of the carotid artery triggers the development of atherosclerosis. Three therapies based on counteracting the role of leukocytes have been developed and tested in clinical trials: a humanized antibody to the CD11b/CD18 integrin (Hu23F2G or LeukArrest) [144]; the recombinant neutrophil inhibitory factor UK-279 that binds to CD11b/CD18 [148]; and mainly a MAb against ICAM-1 (enlimomab, R6.5). Unfortunately, the outcome of these clinical trials was not what was desired, owing to side effects or a lack of efficacy that limited their clinical translation.

Enlimomab is a murine IgG2a MAb directed against extracellular domain 2 of human ICAM-1 [149]. Previous preclinical studies showed that enlimomab prevented brain damage [150]. Unfortunately, in the enlimomab acute stroke trial [73], enlimomab administration within 6 h of stroke onset did not benefit patients.

### 4.3. Ischemic Tolerance and Immunomodulation

Another alternative strategy related to the counteraction of inflammation is immunomodulation. The aim of this strategy is the suppression of the deleterious effects of inflammation while improving its protective potential. Ischemic tolerance is one of the approaches to achieving immunomodulation. 

Ischemic tolerance consists of preconditioning an organ, such as the brain, with a subthreshold level of pathologic stimulus so it acquires protection. Clinical evidence indicates that preconditioning may occur naturally after transient ischemic attacks and mild strokes in humans [151]. It has been demonstrated that a short nondamaging ischemic insult protects the brain from a subsequent damaging ischemic stimulus [152,153]. Likewise, endotoxin tolerance induced by low doses of lipopolysaccharide protects the brain from subsequent ischemic damage [154]. The molecular mechanisms involved in endotoxin tolerance are closely parallel to ischemic tolerance. There is a modulation of the balance of pro-and anti-inflammatory signaling, leading eventually to the shift of toll-like receptors (TLR) 4 signaling the release of IFN-β, which suppresses the induction of inflammatory cytokines and the recruitment of inflammatory cells. 

Depending on the preconditioning stimulus and the conditions, tolerance can be (1) rapid (within minutes), induced by disruption of lipid rafts, changes in membrane microdomains mediating receptor-induced signaling pathways and leading to the inhibition of TLR/cytokine pathways; and (2) a delayed form of tolerance, in which protection is acquired after several hours or days, as de novo protein synthesis is required. In the latter, the preconditioning stimulus first activates the TLR/cytokine inflammatory pathways, triggering both inflammation and upregulation of the feedback inhibitors of inflammation (signaling inhibitors, decoy receptors, and anti-inflammatory cytokines) [155]. Cerebral preconditioning counteracts the acute inflammatory response that exacerbates ischemic brain injury. However, preconditioning seems to be more suitable for stroke prevention in high-risk patients than in acute stroke patients, since “tolerance” needs to be induced prior to injury [156]. In this context, it would also be useful before surgical procedures [155]. 

Other strategies focused on immunomodulation are orientated to shifting the immune response from a Th1- to a Th2-type response. One strategy is exposure to myelin antigens or E-selectin, which promotes a protective Th2 response [157]. The administration of recombinant T-cell receptor ligands also suppresses the infiltration of inflammatory cells and provides neuroprotection [158]. Finally, the induction of the protective roles of Treg has been proposed [159]. 

Although the decreased inflammatory response to ischemic injury may be beneficial in the acute phase of the stroke, it may compromise repair counteracting mechanisms and worsen infectious complications and long-term outcomes [156]. In addition, all these immunomodulatory strategies are in a very preliminary phase. Further studies are necessary to explore the consequences and the short- and long-term benefits of stroke prevention.

## 5. miRNAs as Regulators of Atherosclerosis and Ischemic Stroke

MicroRNAs (miRNAs), a class of noncoding RNA, have emerged as critical regulators of gene expression, acting predominantly at the posttranscriptional level. This large family of short (22-nucleotide) noncoding RNA binds to the 3′ untranslated (3′UTR) region of mRNA, thereby repressing gene expression. Since their discovery in 1998, thousands of miRNAs have been described in both healthy and pathological states. The implication of diverse miRNAs during the different processes of atheroma plaque formation has been shown during the last two decades. It is well known and has been extensively described, how and which miRNAs regulate vascular macrophage, endothelial, and smooth muscle cell dysfunctions during atherosclerotic plaque formation in preclinical models [160,161,162,163,164,165,166], although this is not the focus of the present section. 

Interestingly, there is a large number of clinical studies that relate circulating miRNA expression profiles to ischemic stroke, although from different approaches. For example, several works study the presence of miRNAs in carotid arteries [167,168,169,170]. Others are longitudinal studies that compare the miRNA expression profile between patients with an incident stroke (ischemic, hemorrhagic, or unspecified) vs. no stroke [171,172]. However, this section of the review is focused on those studies that compare the levels of different circulating miRNAs between asymptomatic ICA stenosis (ICAS) patients and healthy controls, as shown in Table 2. Some of these miRNAs differentially expressed in the circulation might be used as biomarkers of the progression of the diseases leading to ischemic stroke and their levels may help to identify patients with high risk of ischemic stroke who are eligible for carotid endarterectomy. Specifically, serum expression levels of miR-106b-5p [173], miR-92a [174], miR-19a-3p [175], miR-483-5p [176], miR-186-5p [177], miR-27b [178], and miR-342-5p [179] were found to be upregulated in asymptomatic ICAS patients versus controls, showing relatively high sensitivity and specificity in differentiating them from healthy subjects. Additionally, logistic regression analyses revealed an association between these miRNAs and the patients’ degree of carotid stenosis, as well as other vascular risk factors, such as diabetes, hypertension, and dyslipidemia. This evidence may provide some support for the involvement of these miRNAs in the development of carotid atherosclerosis.

To further evaluate the predictive value of miRNAs for the occurrence of cerebral ischemic stroke, the enrolled asymptomatic patients were followed up for 5 years. The primary endpoint was the occurrence of ipsilateral ischemic stroke, TIA, or sudden death. The results demonstrated that high levels of miR-106b-5p, miR-92a, miR-19a-3p, miR-483-5p, miR-186-5p, miR-27b, and miR-342-5p in serum could be used as independent predictive factors associated with the risk of the future onset of cerebrovascular events in this kind of patient [173,174,175,176,177,178,179]. Additionally, in another study using peripheral blood exosomes from patients with plaque progression, Dolz et al. stated that the expression of miR-199-3b, miR-27b-3, miR-130a-3p, miR-221-3p, and miR-24-3p was upregulated [180].

In line with this, similar studies have determined that the levels of serum miR-206 [181], miR-9-5p [182], miR-637 [183], miR-503-5p [184], and miR-486-5p [185] were significantly reduced in asymptomatic ICAS patients compared with those in healthy individuals. Their diagnostic accuracy for the patients was high. Furthermore, the downregulation of miR-206, miR-9-5p, and miR-637 in the patients had predictive value for the incidence of cerebrovascular events within 5 years [181,182,183]. Interestingly, an in vitro proliferation assay indicated that overexpression of miR-503-5p significantly inhibited the proliferation of VSMCs, thus improving atherosclerosis [184] and that miR-486-5p prevented endothelial dysfunction in association with an anti-inflammatory and antioxidative effect by targeting the nuclear factor of activated T cells 5 (NFAT5) [185]. 

Thus, it is clear from the number of studies published, that there is a high level of interest in deciphering the use of miRNAs as biomarkers or therapeutic targets for stroke, but it is also true that the design of these studies has some limitations that need to be addressed—for instance, further investigations with a larger study population are needed to confirm the role of miRNAs; some studies did not include healthy controls; and during the follow-up period, lifestyle and other factors of patients were not monitored, leading to poor prognosis. However, all these studies suggest the role of specific circulating miRNA expression profiles as a noninvasive biomarker of carotid plaque. In this regard, the identification of specific miRNAs is essential for developing novel diagnostic and therapeutic tools and strategies, as will be discussed in the next section. 

**Table 2 ijms-24-14325-t002:** Circulating miRNAs have been reported to have a high diagnostic value in identifying asymptomatic ICAS patients from healthy controls and predicting the occurrence of cerebral ischemic events.

miRNA	Study Population	Up- or Downregulation	Diagnostic Value: ROC Analysis	Predictive Value: Cox Regression Analysis (95% CI)	Significant Correlations	Ref.
miR-106b-5p	58 patients vs. 61 controls	↑	AUC 0.911 SE 89.7%, SP 83.6% (cutoff value 0.198)	HR 5.431 CI (1.592–18.520) *p* = 0.007	Dyslipidemia, hypertension, and carotid stenosis	[173]
miR-92a	122 patients vs. 62 controls	↑	AUC 0.895 SE 88.5%, SP 79% (cutoff value 1.285)	HR 2.971 CI (1.230–7.173) *p* = 0.015	Fasting blood glucose, TC, hypertension, and carotid stenosis	[174]
miR-19a-3p	101 patients vs. 98 controls	↑	AUC 0.905 SE 80.2%, SP 86.7%	HR 8.507 CI (2.239–32.328) *p* = 0.002	Carotid stenosis	[175]
miR-483-5p	128 patients vs. 76 controls	↑	AUC 0.910 SE 80.5%, SP 89.5% (cutoff value 0.705)	HR 2.670 CI (1.099–6.484) *p* = 0.030	Diabetes, dyslipidemia, and carotid stenosis	[176]
miR-186-5p	67 patients vs. 60 controls	↑	AUC 0.919 SE 89.6%, SP 81.7% (cutoff value 1.221)	HR 4.190 CI (1.166–15.061) *p* = 0.028	Dyslipidemia, hypertension, and carotid stenosis	[177]
miR-27b	71 patients vs. 58 controls	↑	AUC 0.902 SE 77.5%, SP 94.8% (cutoff value 1.491)	HR 5.067 CI (1.170–21.943) *p* = 0.030	TC, hypertension, and carotid stenosis	[178]
miR-342-5p	92 patients vs. 86 controls	↑	AUC 0.905 SE 85.9%, SP 80.2%	HR 5.512 CI (1.370–22.176) *p* = 0.016	Serum IL-6 and TNFα inflammatory factors	[179]
miR-206	105 patients vs. 101 controls	↓	AUC 0.939 SE 86.7%, SP 86.14% (cutoff value 0.754)	HR 0.046 CI (0.005–0.431) *p* = 0.007	Carotid stenosis	[181]
miR-9-5p	88 patients vs. 86 controls	↓	AUC 0.910 SE 80.7%, SP 87.2% (cutoff value 0.72)	HR 0.239 CI (0.087–0.652) *p* = 0.005	Hypertension	[182]
miR-637	97 patients vs. 90 controls	↓	AUC 0.919 SE 85.6%, SP 83.3% (cutoff value 0.759)	HR 0.073 CI (0.017–0.313) *p* ≤ 0.001	Carotid stenosis	[183]
miR-503-5p	62 patients vs. 60 controls	↓	AUC 0.817 SE 83.3%, SP 79.03% (cutoff value 0.810)	N/A	Diabetes and arterial stenosis	[184]
miR-486-5p	91 patients vs. 87 controls	↓	AUC 0.921 SE 82.4%, SP 89.7% (cutoff value 0.692)	N/A	Carotid stenosis	[185]

Abbreviations: AUC, area under curve; CI, confidence interval; HR, hazard ratio; IL-6, interleukin-6; N/A, not applicable; ROC, receiver operating characteristic; SE, sensitivity; SP, specificity; TC, total cholesterol; TNFα, tumor necrosis factor alpha.

## 6. Putative Strategies Based on miRNA

In preclinical studies, therapies for regulating the activity of miRNAs, inhibiting or overexpressing them, are typically used. Thanks to the development of nanotechnologies, they are emerging as the next frontier in treatment options for different pathologies, such as cancer and atherosclerosis. However, the translation of these miRNA-based therapies into clinical practice has been hampered by different issues associated with the specific delivery, their tolerability, efficacy, and specificity. These issues are the reason why most clinical trials have been terminated. A few clinical trials, the majority in the field of cancer, had positive results. A first-in-human Phase I study assessed the maximum tolerance dose, pharmacokinetics, safety, and clinical activity of a liposomal miR-34a mimic (MRX34) in patients with advanced solid tumors [186], but it was closed down early due to adverse reactions in patients [187]. The MesomiR-1 is another Phase I clinical trial for malignant pleural mesothelioma treatment in which patients were treated with miR-16 mimic [188]. Related to the CV field, the first clinical trial in heart failure patients using CDR132L to target miRN-132-3p (NCT04045405) was recently described. This Phase 1b clinical study was well tolerated and safe, and the pharmacodynamic findings were encouraging. Interestingly, levels of miR-132 in the plasma of patients were reduced and there were some functional cardiac improvements. However, one limitation of this study was the small number of patients [165]. 

Nine ongoing clinical trials are using miRNA-based therapies but related to other diseases, such as type II diabetes and nonalcoholic fatty liver disease (NCT03225846 and NCT04617860), hepatitis C virus infection (NCT01646489, NCT01727934, NCT01872936, NCT01200420), the treatment of keloid (pathological fibrosis) (NCT02603224 and NCT03601052), or Huntington’s disease (NCT04120493). It is clear that further investigations in nonhuman primates and future clinical studies are needed to overcome the challenges of translational miRNA-based therapies into clinical applications for the prevention of stroke or the prospective use of miRNA modulation in carotid atherosclerosis.

## 7. Other Putative and Complementary Strategies

Apart from the anti-inflammatory therapies described and novel therapies based on miRNA, other strategies include lipid-lowering therapies beyond statins, antibiotic drugs, and natural/nutritional medicine.

Lipids play a pivotal role in atherothrombotic stroke and they are potentially important targets, as previously discussed. Therefore, the use of any targeted anti-inflammatory agent should be considered in parallel with statin therapy and/or novel lipid-lowering therapies. Owing to the well-described interactions between lipids and innate immunity, it is very feasible that the overall clinical benefit deriving from intensive combination therapy will be enhanced compared with individual therapy, as proposed by Ridker 2020 [119]. 

Besides statins, other lipid-lowering agents show anti-inflammatory properties [189]. Ezetimibe therapy, which blocks the intestinal absorption of cholesterol, not only lowers LDLc but also enhances the reduction in hsCRP promoted by statin therapy. Importantly, in the FOURIER study, in patients with established atherosclerosis treated with statins, the inhibition of PCSK9, involved in increasing the expression of the LDL receptor, with evolocumab reduced the risk of ischemic stroke [190]. In addition, two of the newest lipid-lowering medications for the prevention and treatment of cardiovascular disease that are in Phase III clinical trials are bempedoic acid and inclisiran [191]. Other future therapies aiming to reduce the modification of LDL and/or to improve the qualitative properties of LDL and HDL may be very valuable in preventing carotid atherosclerosis progression and complications. In this regard, the putative use of apolipoprotein (apo)-based mimetic peptides to reduce atherosclerosis [192] is outstanding. 

It has been hypothesized that some chronic bacterial infections may be associated with the development of atherosclerosis and ensuing complications, such as myocardial infarction and stroke. These bacterial infections seem to play a role in the initiation, progression, and destabilization of atherosclerotic plaques. Supporting epidemiological studies reported the presence of Chlamydia pneumoniae within carotid plaque and associations between chronic infection with C. pneumoniae and stroke risk [193,194]. Also, chronic periodontitis has been linked to carotid plaque destabilization. In this context, sub-antimicrobial dosages of antibiotics have demonstrated benefits in inhibiting inflammation. A study by Sander et al., showed a protective effect of roxithromycin treatment on the progression of arteriosclerosis [195]. Once again, a better knowledge of plasmatic biomarkers associated with chronic infection/immune response will probably help in selecting patients who are candidates for receiving antibiotic drugs.

Recently, increasing interest in the use of natural medicines for stroke has emerged. Some natural compounds seem to elicit a positive effect on microcirculation in the brain by reducing oxidative stress and inflammation. Many of those studies are conducted in animal models of ischemic stroke, particularly in the acute phase of cerebral ischemia, as extensively reviewed by Tao et al. [196]. Several natural medicines are antioxidants that regulate oxidative stress-related signaling pathways and, also, exert anti-inflammatory effects. Therefore, they may be useful as a preventive therapy, particularly in ischemic stroke associated with carotid atherosclerosis. Moreover, many stroke patients are elderly and with comorbidities, such as hypertension and diabetes. In this regard, some studies have shown the therapeutic effect of natural medicine on stroke and its comorbidities, such as a study revealing that Chinese herbal medicine may reduce the risk of stroke in patients with Parkinson’s disease [197]. However, natural medicine has limitations, mainly differences between formulations and batches of the preparations, and the lack of success in its translation into clinical settings. The latter is likely because the preclinical studies are usually performed in healthy young adult rodents whereas stroke predominantly occurs in the aging population.

A basic potential strategy for primary prevention is based on nutrition and physical exercise, whose role in cardiovascular disease and stroke has been widely reviewed previously [198,199]; thus, they will not be discussed in the present review. 

The intake of probiotics and prebiotics also seems to play an important role in preventing and delaying the development of CV disease, an effect partly mediated through the modulation of the levels of LDLc and hsCRP. The underlying mechanisms of their protective effect emerge from promoting changes in gut microbiota and modulating inflammatory responses [200]. 

## 8. Conclusions and Future Investigations

In summary, several anti-inflammatory strategies have been suggested as therapies in the prevention of ischemic stroke, particularly of the atherothrombotic subtype, where inflammation plays a major role. Broad-spectrum anti-inflammatory therapies used long-term in low doses are well-known candidates. In this context, colchicine has been the focus of several clinical studies with controversial results, mainly due to a lack of specificity and ensuing deleterious collateral effects. On the other hand, other therapies have been directed against specific inflammatory molecules (cytokines, adhesion molecules) by using monoclonal antibodies. Although these strategies showed promising results in preclinical studies with animal models, they failed to prevent ischemic stroke in clinical trials. Part of the lack of success is likely due to the complexity of the immune response. In addition, animal models do not replicate exactly the complexity of pathologies, such as stroke, especially if the studies are conducted in young animals. 

Other interesting and promising therapies based on counteracting inflammation have emerged. Some of them are based on promoting immunomodulation or the use of miRNA mimics. They are in a much more preliminary stage, a fact that hampers their implementation in clinical practice but deserves future investigations. 

Therefore, although some anti-inflammatory and immunomodulatory therapies have been suggested for the prevention of atherothrombotic ischemic stroke, they are far from being available for clinical use. To choose the best candidate to target stroke, future studies addressing the inflammatory/immune mechanisms leading to vulnerable carotid plaque are essential. Moreover, an appropriate strategy would be targeting not only a single inflammatory cell/molecule but several ones, in combination with lipid-lowering therapies. Future studies should take into account the specific stroke subtypes, considering particularly ischemic stroke associated with carotid atherosclerosis, in which anti-inflammatory therapies may show the strongest benefit. 

Although several limitations hamper the achievement of an effective target for therapy, the protective effects of anti-inflammatory strategies in preclinical models justify additional investigations to achieve a successful clinical translation. In this context, further studies are necessary to ascertain which patients may benefit from anti-inflammatory agents and how.

## Figures and Tables

**Figure 1 ijms-24-14325-f001:**
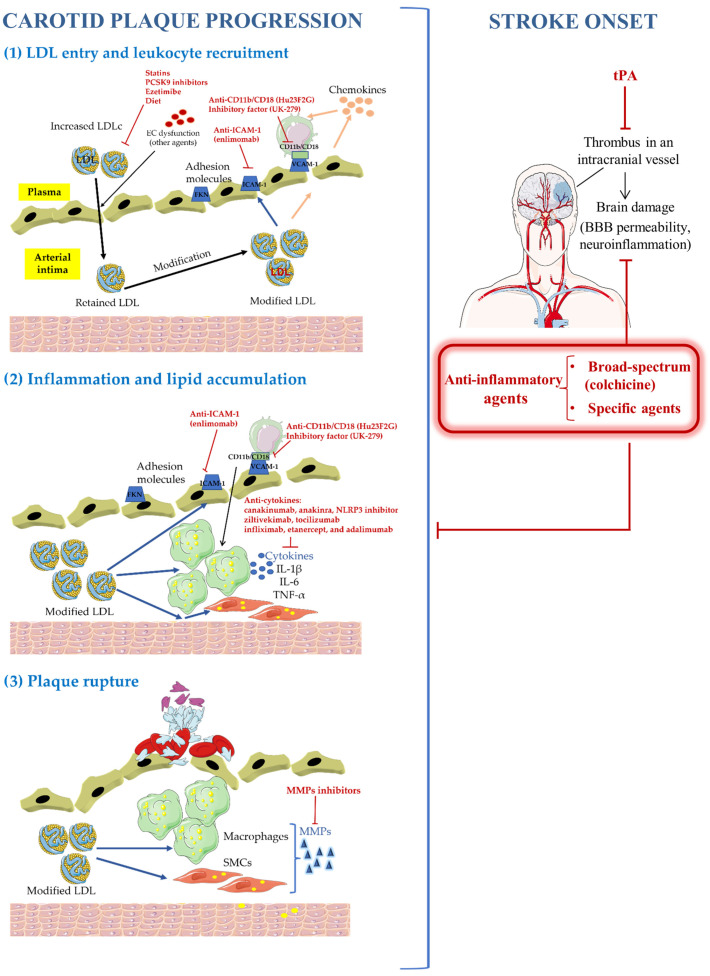
Anti-inflammatory strategies based on slowing the progression of atherosclerosis to prevent ischemic stroke. tPA: tissue plasminogen activator.

**Table 1 ijms-24-14325-t001:** Clinical trials involving anti-inflammatory therapies that include stroke among clinical outcomes.

Drug	Clinical Trial	Sample Size and Study Population	Design	Intervention	Outcome	Effect
Broad-spectrum anti-inflammatory drugs						
Colchicine	LoDoCo	*n* = 532, patients with stable CAD	Single-center, randomized, observer-blinded	Colchicine 0.5 mg/day vs. control	Acute coronary syndrome, out-of-hospital cardiac arrest, or noncardioembolic ischemic stroke	Effective for the prevention of CV events
	LoDoCo2	*n* = 5522, patients with chronic CAD	Multicenter, randomized, double-blind	Colchicine 0.5 mg vs. placebo	CV death, spontaneous MI, ischemic stroke	Effective for the prevention of CV events
	COLCOT	*n* = 4745, patients with CAD randomized within the first 30 days after MI	Multicenter, randomized, double-blind	Colchicine 0.5 mg vs. placebo	Death from CV causes, resuscitated cardiac arrest, MI, stroke, coronary revascularization	Effective for the prevention risk of CV events
	COPS	*n* = 795, patients with acute coronary syndrome and CAD	Multicenter, randomized, double-blind	Colchicine 0.5 mg vs. placebo	All-cause mortality, acute coronary syndrome, revascularization, noncardioembolic ischemic stroke	Improved clinical outcome in the colchicine group
	CLEAR SYNERGY	*n* = 7063, patients with MI	Multicenter, randomized, double-blind, double-dummy, 2 × 2 factorial design	Colchicine 0.5 mg vs. placebo or Spironolactone 25 mg vs. placebo)	CV death, recurrent MI, stroke	Ongoing trial
	CONVINCE	*n* = 2623, patients with noncardioembolic ischemic stroke or TIA	Multicenter, randomized, double-blind	Colchicine 0.5 mg/day vs. usual standard of care	Recurrence of vascular events	Ongoing trial
	CASPER	Not yet recruiting, patients with ischemic stroke or TIA and hsCRP ≥ 2 mg/L	Multicenter, randomized, double-blind	Colchicine 0.5 mg/day vs. usual standard of care	Recurrence of vascular events	Ongoing trial
Methotrexate	CIRT	*n* = 4786, patients with MI and type 2 diabetes	Multicenter, randomized, double-blind	Methotrexate 15–20 mg vs. placebo	CV events, all-cause mortality, congestive heart failure	No results in CV events
Minocycline	MINOS	*n* = 60, patients with acute ischemic stroke	Single-center, Open-label, dose escalation	3, 4.5, 6, or 10 mg/kg daily over 72 h	Safety, pharmacokinetic, mRS at 3 months	Safe and neuroprotective
Lp-PLA2 inhibitory drugs						
Varespladib	VISTA-16	*n* = 5145, patients with acute coronary syndrome	Multicenter, randomized, double-blind	Varespladib 500 mg vs. placebo	CV mortality, MI, stroke, angina	No results in recurrent CV events and risk of MI
IL-1β blocking drugs						
Canakinumab	CANTOS	*n* = 10,066, patients with MI and atherosclerosis	Multicenter, randomized, double-blind	Canakinumab 50, 150 or 300 mg vs. placebo	Nonfatal MI, stroke, CV death, inflammatory burden	All doses reduced hsCRP; but only the 150 mg dose reduced nonfatal MI, stroke, or CV death
Anakinra	SCIL-STROKE	*n* = 80, patients presenting within 5 h of ischemic stroke onset	Single-center, randomized, double-blind	Anakinra 100 mg vs. placebo	mRS at 3 months	Reduction in plasma inflammatory markers associated with worse clinical outcome
IL-6 receptors blocking drugs						
	ZEUS	*n* = 6200, patients with chronic kidney disease and hsCRP ≥ 2 mg/L	Multicenter, randomized, double-blind	Ziltivekimab 15 mg vs. Placebo	CV death, nonfatal MI, stroke	Ongoing trial
ICAM-1 inhibitory drugs						
Enlimomab	Enlimomab Acute Stroke Trial	*n* = 625, patients with ischemic stroke	Single-center, randomized, double-blind	Enlimomab 160 mg vs. placebo	mRS at 3 months, NIHSS, survival	Enlimomab worsens stroke outcome

Abbreviations: CAD, coronary artery disease; CV, cardiovascular; hsCRP, high-sensitivity C-reactive protein; ICAM-1, intercellular adhesion molecule-1; IL, interleukin; Lp-PLA2, lipoprotein-associated phospholipase A2; MI, myocardial infarction.

## Data Availability

No new data were created in this study. Data sharing is not applicable to this article.

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
