# Peer review of "Novel Therapeutic Approaches to Prevent Atherothrombotic Ischemic Stroke in Patients with Carotid Atherosclerosis"

_ijms, 2023, doi:10.3390/ijms241814325_

Round 1

Reviewer 1 Report

This is a well written comprehensive review thoroughly discusses existing anti-inflammatory therapies but also highlights novel strategies addressed to counteract inflammation and their potential clinical use in the prevention of ischemic stroke particularly in the atherothrombotic subtype.

You should provide the appropriate information required (or delete) in the sections: Institutional Review Board Statement, Informed Consent Statement, Data Availability Statement, Acknowledgments, Conflicts of Interest, Appendix A, Appendix B (lines 765-808)

Author Response

We thank the reviewer for his/her kind comments. We have deleted the information in the indicated sections

Reviewer 2 Report

This is a review article on novel therapeutic approaches to prevent atherothrombotic ischemic stroke in patients with carotid atherosclerosis. This is a well-written comprehensive review about the immune and inflammatory processes of atherosclerosis and the corresponding methods of treatment with 197 cited references.

There are several points to be concerned in this article before it is accepted for publication.

 1.     Line 57: “In antiplatelet therapy, aspirin, a non-56 steroidal anti-inflammatory drug that inhibits cyclooxygenase 2 (COX-2)…”. Please reassure the mechanism of aspirin, Cox-1 or Cox-2 inhibitor, or both?

2.     Figure 1: The resolution of figure 1 is not good enough to read. Was it a typo of “intracraneal vessel”?

3.     Some sections were not completed, such as [Supplementary Materials], [Institutional Review Board Statement], [Informed Consent Statement], [Data Availability Statement], [Acknowledgments], [Acknowledgments], [Appendix A] and [Appendix B].

Author Response

  1. Line 57: “In antiplatelet therapy, aspirin, a non-56 steroidal anti-inflammatory drug that inhibits cyclooxygenase 2 (COX-2)…”. Please reassure the mechanism of aspirin, Cox-1 or Cox-2 inhibitor, or both?

We thank the reviewer for the comment. In fact, aspirin promotes acetylation of both enzymes, Cox-1 and Cox-2. Now, the sentence has been corrected.

  1. Figure 1: The resolution of figure 1 is not good enough to read. Was it a typo of “intracraneal vessel”?

We have improved the resolution of Figure 1. We hope that now it is clearer than before.

The indicated word in the Figure has been corrected.

  1. Some sections were not completed, such as [Supplementary Materials], [Institutional Review Board Statement], [Informed Consent Statement], [Data Availability Statement], [Acknowledgments], [Acknowledgments], [Appendix A] and [Appendix B].

Thank you for your comment. We have deleted the information in the indicated sections, because it was not applicable to our article.

Reviewer 3 Report

1.  The first line of the abstract should be deleted as it is mentioned again in the literature review.

2. The literature review fails to mention several computational/numerical works that have addressed hemodynamics as it related to stoke and ischemic transient attack.  Please revise to include the following:

Hewlin, R.L., Jr.; Kizito, J.P. Evaluation of the Effect of Simplified and Patient-Specific Arterial Geometry On Hemodynamic Flow In Stenosed Carotid Bifurcation Arteries. Proc. ASME Early Career Tech. J. 2011, 10, 39–44

Hewlin, R.L., Jr.; Tindall, J.M. Computational Assessment of Magnetic Nanoparticle Targeting Efficiency in a Simplified Circle of Willis Arterial Model. Int. J. Mol. Sci. 202324, 2545. https://doi.org/10.3390/ijms24032545

Li, G., Wang, H., Zhang, M. et al. Prediction of 3D Cardiovascular hemodynamics before and after coronary artery bypass surgery via deep learning. Commun Biol 4, 99 (2021). https://doi.org/10.1038/s42003-020-01638-1

also include invitro experimental works

3. Figure 1 needs a subcaption.

4. The conclusion should be revised to bullet point the significance of each type of work, gaps in knowledge and future work that should be done.

5. The manuscript needs to be formatted to fit the journal guidelines.  Tables should not run over pages.  There is significant white space as well.

Author Response

  1. The first line of the abstract should be deleted as it is mentioned again in the literature review.

Following the reviewer’s recommendation, the first sentence has been deleted

  1. The literature review fails to mention several computational/numerical works that have addressed hemodynamics as it related to stoke and ischemic transient attack. Please revise to include the following:

Hewlin, R.L., Jr.; Kizito, J.P. Evaluation of the Effect of Simplified and Patient-Specific Arterial Geometry On Hemodynamic Flow In Stenosed Carotid Bifurcation Arteries. Proc. ASME Early Career Tech. J. 2011, 10, 39–44

Hewlin, R.L., Jr.; Tindall, J.M. Computational Assessment of Magnetic Nanoparticle Targeting Efficiency in a Simplified Circle of Willis Arterial Model. Int. J. Mol. Sci. 2023, 24, 2545. https://doi.org/10.3390/ijms24032545

Li, G., Wang, H., Zhang, M. et al. Prediction of 3D Cardiovascular hemodynamics before and after coronary artery bypass surgery via deep learning. Commun Biol 4, 99 (2021). https://doi.org/10.1038/s42003-020-01638-1

also include invitro experimental Works

Thank you for your comment. We agree with the reviewer that haemodynamics is a cause of atherothrombotic ischemic stroke that deserves to be mentioned (lines 47-49). In this regard, as indicated in the references kindly provided by the reviewer, computational models based on artery geometry are useful to predict haemodynamics triggering plaque formation and stenosis degree progression. This information has also been added now to the manuscript (lines 137-141).

 3.Figure 1 needs a subcaption.

We have improved Figure 1, by increasing the resolution and highlighting the 3 subcaptations of the steps involved in the progression of atherosclerosis. If the term subcaptation refers to a different issue, please specify it.

  1. The conclusion should be revised to bullet point the significance of each type of work, gaps in knowledge and future work that should be done.

We have rewritten the whole section “Conclusions and future investigations” according to the suggestions provided by the reviewer.

  1. The manuscript needs to be formatted to fit the journal guidelines. Tables should not run over pages. There is significant white space as well.

We have tried to improve this point. Table 1 and Table 2 have been reduced.

Round 2

Reviewer 3 Report

Please include all suggested references.

Author Response

We thank the reviewer for the suggestion.  All the references proposed have been included